



# The Climate Impact of COVID19 Induced Contrail Changes

Andrew Gettelman[1], Chieh-Chieh Chen[1], and Charles G. Bardeen[1]

[1]National Center for Atmospheric Research, Boulder, CO, USA

**Correspondence:** Andrew Gettelman (andrew@ucar.edu)

**Abstract.** The COVID19 pandemic caused significant economic disruption in 2020 and severely impacted air traffic. We use a state of the art Earth System Model and ensembles of tightly constrained simulations to evaluate the effect of the reductions in aviation traffic on contrail radiative forcing and climate in 2020. In the absence of any COVID19 pandemic caused reductions, the model simulates a contrail Effective Radiative Forcing (ERF) of $62\pm59$ mWm$^{-2}$ (2 standard deviations). The contrail

ERF has complex spatial and seasonal patterns that combine the offsetting effect of shortwave (solar) cooling and longwave (infrared) heating from contrails and contrail cirrus. Cooling is larger in June–August due to the preponderance of aviation in the N. Hemisphere, while warming occurs throughout the year. The spatial and seasonal forcing variations also map onto surface temperature variations. The net land surface temperature change due to contrails in a normal year is estimated at $0.13\pm0.04$ K (2 standard deviations) with some regions warming as much as 0.7K. The effect of COVID19 reductions in flight

traffic decreased contrails. The unique timing of such reductions, which were maximum in N. Hemisphere spring and summer when the largest contrail cooling occurs, means that cooling due to fewer contrails in boreal spring and fall was offset by warming due to fewer contrails in boreal summer to give no significant annual averaged ERF from contrail changes in 2020. Despite no net significant global ERF, because of the spatial and seasonal timing of contrail ERF, some land regions that would have cooled slightly (minimum -0.2K) but significantly from contrail changes in 2020. The implications for future climate

impacts of contrails are discussed.

## 1   Introduction

COVID19 pandemic lockdowns caused lots of economic disruption in 2020. The reduction in Greenhouse Gases (GHGs) and pollution (Le Quéré et al., 2020) likely impacted global temperatures (Forster et al., 2020). GHG reductions would have resulted in cooling and aerosol reductions would have resulted in warming (Gettelman et al., 2021). One of the hardest hit

sectors was aviation, since it was a prime cause of the rapid spread of the pandemic. Total flights dropped by nearly 70% (Figure 1) during the height of lockdowns in spring 2020, and had still not recovered to their pre-pandemic levels by the end of 2020.

   Aircraft have many environmental impacts, including climate impacts. As recently reviewed by Lee et al. (2021), global aviation warms the planet through both $CO_2$ and non-$CO_2$ contributions. Global aviation contributes 3.5% to total anthro-

pogenic radiative forcing, but non-$CO_2$ effects comprise about 2/3 of the net radiative forcing. The largest single contribution





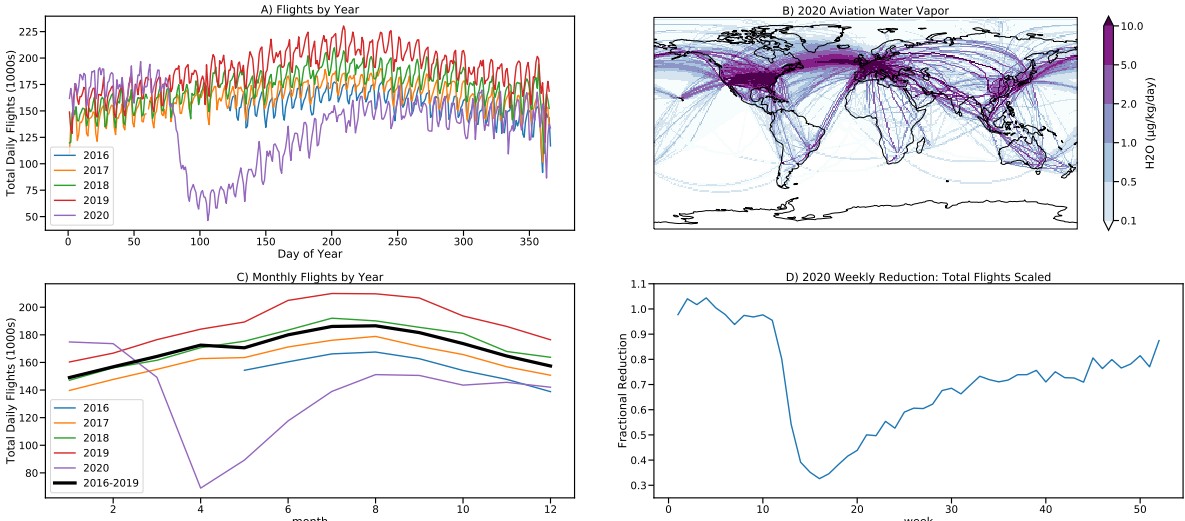

**Figure 1.** A) Daily total flights from 2016–2020, B) Map of 2020 flight level water vapor emissions ($\mu g$ kg$^{-1}$ day$^{-1}$), C) Monthly average flights each year and 2016-2019 average (thick black line), D) Scaled weekly estimate of COVID19 affected flight fraction for 2020.

to aviation radiative forcing is contrails and contrail cirrus, with an estimated 2018 impact of 0.06 Wm$^{-2}$ (60mWm$^{-2}$), with high uncertainty.

The 2020 changes in air traffic likely resulted in reductions in contrail frequency (Schumann et al., 2021). Aircraft contrails create linear condensation trails that can evolve and persist in supersaturated air as contrail cirrus. Like other cirrus clouds, the
resulting clouds scatter solar (shortwave, SW) radiation back to space, cooling the planet. Contrails also absorb and re-emit infrared (longwave, LW) radiation from the earth at colder temperatures, warming the planet. The net effect depends on the cirrus microphysical and radiative properties, and the variation of SW radiation. Integrating over space and time, the net effect of contrails is to warm the planet (Lee et al., 2021) through a balance of the longwave (warming) and shortwave (cooling). Thus reductions in contrails due to COVID19 would be expected to cool the planet.
This study will document the updated version of the contrail model that is publicly available as part of the Community Earth System Model version 2.2 and it's contrail ERF for a 'normal' year 2020 aviation with no pandemic reductions. We then use estimates of the changes in aviation emissions for the full year of 2020 to estimate changes in the contrail Effective Radiative Forcing (ERF) due to the COVID19 pandemic. ERF includes fast temperature adjustments due to changes in cloud formation, and is the usual metric for understanding and assessing changes to the earth's radiation budget.
Section 2 contains the methodology of the model and simulations, Section 3 contains results of the simulations and Conclusions are in Section 4.



## 2 Model and Methods

### 2.1 Model

Simulations use the Community Earth System Model version 2 (CESM2), (Danabasoglu et al., 2020). The atmospheric model
in CESM is the Community Atmosphere Model version 6.2 (CAM6) Gettelman et al. (2020). CAM6 uses a detailed 2-moment
cloud microphysics scheme (Gettelman and Morrison, 2015) coupled to an aerosol microphysics and chemistry model (Liu
et al., 2016), as described by Gettelman et al. (2019).

To the standard version of CESM, we add the contrail parameterization of Chen et al. (2012) that was developed for CAM5.
Since the ice cloud microphysics and aerosols are not substantially different between CAM5 and CAM6, the translation is
straightforward. As described by Lee et al. (2021), we adjust the assumed emission ice particle diameter to 7.5 microns from the
original parameterization (10 microns), to better align with observations. The representation of contrails in CESM, described
by Chen et al. (2012) and Chen and Gettelman (2013), adds aviation emissions of water vapor. A specified mass of water vapor
is emitted based on a data set of total aircraft distance traveled, assuming a contrail diameter of 100m over the grid box length
and standard emission indicies for commercial aircraft (see Chen et al. (2012) for details). If the temperature and humidity
conditions indicate contrail formation (Appleman, 1953; Schumann, 1996), then the vapor is converted to ice crystals with
a specified initial particle size of 7.5 microns (assuming small spherical ice crystals), yielding an ice number concentration
increase. If conditions do not imply contrail formation, then water vapor is added. The vapor or ice is then is part of the fully
conservative CESM hydrologic cycle with all microphysical processes active. When ice is formed, a 100m wide cloud is added
to the cloud fraction (for 100km horizontal resolution, the cloud fraction added is thus 1.e-4 per aircraft). The model can thus
simulate linear contrails (representing a small cloud fraction in a model grid box), as well as their evolution into contrail cirrus,
and their effect on the background environment and existing clouds. For this study we focus only on the impact of aviation
water vapor emissions. Aviation aerosols may have substantial effects on clouds (Gettelman and Chen, 2013), but are highly
uncertain (Lee et al., 2021), and we will focus solely on the effects of water vapor.

For CESM we use the standard 32 levels (to 3hPa) vertical and ∼1° horizontal resolution. Winds are nudged as described
by Gettelman et al. (2020) and Gettelman et al. (2021). The model timestep is 1800 seconds. Winds, Sea Surface Temperatures
(SST) and optionally temperatures are relaxed to NASA Modern-Era Retrospective analysis for Research and Applications,
version 2 (MERRA2) (Molod et al., 2015), available every 3 hours. The linear relaxation time is 24 hours. CESM2 has a fully
interactive land surface model (the Community Land Model version 5, Danabasoglu et al. (2020)).

These simulations permit temperature adjustment. Resulting radiative flux perturbations constitute an Effective Radiative
Forcing (ERF). We also conduct sensitivity tests as discussed below where temperatures are nudged to MERRA2.

### 2.2 Emissions Data

To simulate aviation emissions for 2020 with and without COVID changes, we modify existing aviation inventories used
with CESM. We take the Aviation Climate Change Research Initiative (ACCRI) 2006 inventory (Wilkerson et al., 2010)
used with earlier versions of CESM (CAM5) (Chen et al., 2012; Gettelman and Chen, 2013). We focus only on water vapor





emissions and contrails. We do not consider effects of aviation aerosols in this study. The ACCRI 2006 inventory was developed
      based on detailed flight track data. The distribution of flight level water vapor emissions is shown in Figure 1B. We make the
      assumption that air traffic has increased significantly since 2006, but that the flight locations and relative density have not
      changed drastically. In some rapidly developing regions of the planet such as China, this assumption will result in some
      additional uncertainty.

80        To estimate the 2020 emissions we estimate the growth in fuel use since 2006 as equal to the growth in total aircraft distance
      traveled using data from Lee et al. (2021). Lee et al. (2021) report 54.7 million km of travel in 2018, and 33.2 million km
      in 2006. The scaling from 2006 to 2018 is then 1.58. We evaluate 2018–2020 increases in aviation emissions against aircraft
      movement data for total (commercial, private and military) flights from Flightradar24 from 2016–2020, illustrated in Figure 1.
      This data shows growth over the last few years of 9%/yr. We thus use a 9%/year increase over 2018–2020 (2 years) to generate

a scaling from 2006 to 2020 of 1.88 (88% increase from 2006) in a scenario without any COVID19 induced reduction to
      aviation.

          In order to determine the perturbation due to COVID19 lockdowns, we use daily data for total flights for each day of 2020
      provided by Flightradar24 (available at, https://www.flightradar24.com/data/statistics), illustrated in Figure 1A, and compare
      this to a scaled up average of previous years 2016–2019, which is 9% above 2019 (Figure 1C). We use weekly averages since

there is a strong weekly cycle in flights (Figure 1A). The analysis yields a scaling value for every week of 2020 from our
      reference (scaled up 2006 emissions), as illustrated in Figure 1D. The first few weeks of 2020 were normal, then reductions
      started in February 2020 due to restrictions in China and Asia, and then in March (around week 12), most nations began
      lockdowns and most commercial flights were halted. Total aviation declined by 2/3 from what would have been expected.
      Recovery was rapid for about 10–15 weeks until the middle of 2020, and then recovery has slowed, reaching approximately

75–80% of the expected value by the end of 2020. Note that this is total flights, including commercial (passenger and cargo),
      private and military (with transponders). The total load factor on passenger flights has decreased, so the total passenger miles
      flown is different than this. But it is total flights that is most relevant for water vapor emissions.

          We then have a scaling factor for 2020 from 2006 (1.88) and weekly modifications to that factor for COVID19 impacted
      emissions. These aviation water vapor emissions are used in our simulations to initiate contrails. All other emissions come

from the Shared Socioeconomic Pathway (SSP) 245 emissions for 2019–2020 and are the same for all simulations.

### 2.3   Simulations

Full aviation simulations with 10 ensemble members are launched from January 1, 2019 to December 31, 2020 (2 years),
with a small temperature perturbation ($10^{-10}$K). The initial perturbation results in a slightly different atmosphere evolution
for each ensemble member. Nudging keeps the atmosphere in a similar 'weather' state. The perturbation samples random

fluctuations within that state. Critically, this enables estimates of the statistical significance of differences. We compared 10
      and 20 ensemble members, and found that 10 members did not change the standard deviation and significance levels for full
      aviation emissions. We define statistical significance for maps with the False Discovery Rate (FDR) method of Wilks (2006),
      which reduces patterned noise. We use the standard deviation across the ensembles to estimate uncertainty and variability





for global averaged quantities. A similar methodology was used to examine non-aviation COVID19 related aerosol emissions
perturbations by Gettelman et al. (2021).

We run simulations with full aviation water vapor emissions (Full Air) and no aviation water vapor emissions (No Air). We
can analyze 2019 and 2020 effects with different meteorology in the 2 year simulations. As will be noted below, the land surface
takes a few months to react to adjusted forcing (Figure 2G), but the other variables adjust quickly (see section 3.1). We also
run an ensemble of 20 members restarted January 1, 2020 with COVID19 reduced aviation water vapor emissions (COVID)
for 2020. 20 ensemble members are used due to the smaller peturbation. Finally we also run a pair of 2020 ensembles with
temperature nudging (Full Air T Nudge, COVID T Nudge) to explore how the evolution of temperature may affect the results.

## 3   Results

First we analyze global mean results by month in Section 3.1. We focus on the differences between ensembles with and without
aviation or COVID19 affected aviation for key climate parameters. Then we assess the spatial and seasonal distribution of these
parameters (Section 3.2). This puts overall global values in important context for assessing contrail ERF and COVID19 reduc-
tions to contrails. For clarity in dates we will refer to the COVID19 affected aviation simulations in the figures as 'COVID'.
Finally we look in more detail at cloud changes and the effects of temperature nudging on the climate response to aviation
contrails (Section 3.3).

### 3.1   Global Mean Results

Figure 2 illustrates monthly global mean quantities from the simulations. Shaded regions are two standard deviations ($\pm 2\sigma$)
across the ensemble. Global annual mean quantities are provided in Table 1. Figure 2 shows dates in 2020, but also illustrates
the differences in the 2019 spin up year (green solid lines, mapped to the 2020 annual cycle), which has the same aviation
emissions but different meteorology. It is clear that the land surface temperature takes about 4 months to come to equilibrium
with the forcing (Figure 2H), but the other fields are all similar for all months.

Aviation contrails (Full Air - No Air) cause increases in the negative SW Cloud Radiative Effect (CRE), a net cooling (Fig-
ure 2A), and a LW CRE warming (Figure 2B) (green, orange and purple dash). The opposite effects are seen when COVID19
reductions (COVID - Full Air) in contrails are assessed (blue and red dashed). There is an annual cycle in the SW CRE
(Figure 2A) with a peak cooling in Northern Hemisphere (NH) summer, when maximum sunlight occurs in the regions of
maximum emissions at NH mid-latitudes. The LW CRE (Figure 2B) has virtually no annual cycle. The COVID19 emissions
changes should then be noted in the context of this annual cycle. The LW CRE changes due to COVID19 reductions (Fig-
ure 2B, blue solid and red dashed) clearly shows differences that map directly to the temporal evolution of aviation reductions
(Figure 1D). The phase of the SW CRE (Figure 2A) and LW CRE (Figure 2B) for the COVID case do not exactly line up (peak
SW in August, peak LW in April). This is because of the convolution between reductions (Figure 1D) with the peak in the SW
contrail effect (Figure 2A).



**Table 1.** Global Annual Mean Differences of fields shown in Figure 2. Uncertainties are 2 standard deviations across each ensemble.

| Field | units | 2020 Full Air | 2019 Full Air | Full Air T Nudge | COVID | COVID T Nudge |
|---|---|---|---|---|---|---|
| SW CRE | mWm$^{-2}$ | -446±40 | -471±65 | -290±20 | 165±47 | 91±17 |
| LW CRE | mWm$^{-2}$ | 558±22 | 620±28 | 675±19 | -153±24 | -171±17 |
| TOA Flux | mWm$^{-2}$ | 33±35 | 90±50 | 32±16 | 27±58 | -69±18 |
| IWP | gkg$^{-1}$ | 0.36±0.019 | 0.41±0.017 | 0.45±0.010 | -0.086±0.018 | -0.11±0.011 |
| Cld Top Ni | L$^{-1}$ | 0.07±0.26 | 0.26±0.29 | 0.20±0.16 | -0.032±0.27 | -0.150±0.123 |
| Cld Top Rei | microns | -0.52±0.032 | -0.53±0.029 | -0.79±0.013 | 0.083±0.030 | 0.14±0.015 |
| High Cld Fract | Percent | -0.001±0.001 | -0.003±0.006 | -0.27±0.033 | -0.01±0.067 | 0.03±0.035 |
| Surf T Land | K | 0.134±0.04 | 0.116±0.03 | -0.032±0.03 | -0.02±0.004 | 0.004±0.004 |

The Ice Water Path (IWP) due to full contrail effects (Figure 2C) has a small annual cycle, and is similar to LW CRE
(Figure 2B), which is sensitive to ice mass. The ratio of the LW CRE change to IWP change is similar for the 2020 Full Air
(1.6), 2019 Full Air (1.6) and COVID (1.8) ensembles. There is little change relative to the variability in global average cloud
top ice number (Figure 2D), but this masks regional variability to be discussed below (Section 3.2). The average size of ice
crystals decreases due to aviation contrails (Figure 2E) and correspondingly increases when aviation contrails are reduced.
This is expected as contrails add small (7.5 micron initial diameter) ice crystals into the model.

The changes in high cloud (above 400 hPa) fraction are small (Figure 2F). Few of the changes are significantly different
from zero, mostly only in the summer period for full aviation emissions. There is a different annual cycle when temperature
nudging is used (Figure 2F, purple). These global changes mask spatial and vertical structure in cloud field changes we will
analyze in Section 3.3 below.

The Top Of Atmosphere (TOA) radiative flux (Figure 2G) is a residual of positive LW CRE and negative SW CRE, with
potential additional components due to possible effects of clouds on clear sky aerosols and surface temperature changes. In
general the LW CRE dominates: aviation contrails are a net warming effect over the annual cycle (Table 1), assessed at 33 or
90 mWm$^{-2}$ depending on meteorology for 2020 and 2019 respectively. Combining the means and standard deviations yields
an estimate of 62±59 mWm$^{-2}$ (±2$\sigma$ range). This is very similar to the estimate from Lee et al. (2021) for 2018. Note that the
COVID reduced aviation emissions have offsetting LW and SW effects due to the timing of the aviation reductions such that
the net global effect is actually positive but not distinguishable from zero (27±58 mWm$^{-2}$).

Even with these effects, there are very small but potentially significant changes in land surface temperature (Figure 2H).
Note that because of limited land area and seasonal evolution, the land surface temperature changes may differ from net global
TOA radiation changes. Note that these simulations assume zero temperature change over the ocean, and thus do not include
slow ocean feedbacks. However, the observed ocean temperatures are in equilibrium with full aviation effects. The 2019
simulations (Figure 2H, Green) illustrate the equilibration of the land surface takes 4 months or so. After 4 months the 2020
and 2019 results are nearly identical for surface temperature. The atmospheric fields equilibrate much more rapidly, seen in the
similarity between green (2019) and orange (2020) lines in Figure 2A-G. The spatial structure will be assessed below. Thus,



**Figure 2.** Global monthly mean differences between sets of ensembles. Reductions due to COVID19 aviation changes (COVID - Full Air, Blue), 2020 all aviation (Full Air - No Air, Orange), all aviation using 2019 meteorology (Full Air - No Air, Green), COVID19 changes with temperature nudging (COVID - Full Air, red dash) and full aviation with temperature nudging (Full Air - No Air, purple dash) A) Shortwave (SW) Cloud Radiative Effect (CRE), B) Longwave (LW) CRE, C) Ice Water Path, D) Cloud Top ice number concentration, E) Cloud top ice effective radius, F) High cloud fraction, G) Net Top of Atmosphere (TOA) radiation difference and H) Land surface temperature difference. Shading indicates two standard deviations of global means across the ensembles.



aviation contrails cause a significant land surface temperature warming averaged over the 'normal' (no COVID19 reductions)
2020 annual cycle of 0.13±0.04 K. The COVID19 reductions in aviation caused a cooling over land of -0.03±0.03K, even
though there is no significant net TOA radiation difference (Figure 2G and Table 1). This is understandable based on the
patterns of TOA flux, to be discussed below (Section 3.2).

## 3.2   Spatial Patterns

Figure 3 illustrates the annual average spatial distribution of the climate quantities in Figure 2 for the effect of full aviation
contrails. Stippling indicates statistical significance at the 90% level using the FDR methodology (Wilks, 2006). The expected
pattern for many of the climate impacts matches the distribution of aircraft flight tracks (Figure 1B), with peaks over Eastern
North America, Europe and Southeast and East Asia as well as the North Atlantic and North Pacific oceans. A majority of
the effects occur over the Northern Hemisphere. Contrails induce a cooling due to the SW CRE (Figure 3A) and a co-incident
warming due to LW CRE (Figure 3B). This arises due to increases in the IWP (Figure 3C). There are significant regional
increases in ice number concentration (Figure 3D) not evident in the global averages (Figure 2D), concentrated where the IWP
increases. The ice crystal size decreases (Figure 3E) in a more diffuse but monotonic pattern, leading to a more consistent
global decrease (Figure 2E). High cloud fraction has a more complex pattern of increases in the subtropics at flight altitudes,
with decreases at higher latitudes over most of the Northern Hemisphere. This will be examined in more detail in Section 3.3
below with the vertical structure of cloud fraction and IWP changes.

The result of all of these changes is significant increases in TOA radiative flux (Figure 3G) over parts of the Northern
Hemisphere in or adjacent to flight lanes. Note there are some significant remote decreases in TOA flux in regions with
decreasing high clouds and negative LW and positive SW effects. So not only do LW and SW CRE effects of contrails cancel,
but there are spatial regions of increasing and decreasing TOA flux. The resulting TOA fluxes over land lead to increases in
land surface temperature nearly everywhere, peaking in the subtropical regions of Africa and Asia at 0.7K. The pattern is not
dependent on specific meteorology: similar patterns of warming in Western N. America, and Subtropical Africa, the Middle
East and Asia also occur with 2019 meteorology (not shown).

The pattern of warming is due to the seasonal cycle, illustrated in Figure 4. The lack of a significant annual mean warming
signal over Eastern N. America and reduced signal over Europe is due to the seasonal cycle: there is warming in winter
(December–February) over Europe and moderate warming over the U.S., with cooling at higher latitudes. In summer (June–
August) however there is significant cooling over Eastern N. America and N. Europe. This arises from the seasonal cycle of
TOA flux (Figure 2G) which is negative in N. Hemisphere summer due to strong SW CRE cooling from contrails (Figure 2A),
while the LW warming is more constant over the year (Figure 2B). The TOA SW effects land temperatures directly in the
absence of clouds, while the LW is filtered through the atmosphere, hence the TOA net radiation affects the surface differently
in different seasons.

The changes due to COVID19 reductions in contrails (Figure 5) are as expected: smaller and of the opposite sign to the full
contrail effect (Figure 3) as contrails are reduced. The contour intervals in Figure 5 are smaller than Figure 3, so some of the
ensemble variability (noise) shows up, especially in the tropics (e.g. Figure 5G). In general similar patterns are seen in SW

**Figure 3.** Annual mean maps of differences for Full air - No air for 2020. A) Shortwave (SW) Cloud Radiative Effect (CRE), B) Longwave (LW) CRE, C) Ice Water Path, D) Cloud top ice number concentration, E) Cloud top ice effective radius, F) High cloud fraction, G) Net Top of Atmosphere (TOA) radiation difference and H) Land surface temperature difference. Stippled regions are significant differences using an FDR test at 90% confidence.

**Figure 4.** Seasonal mean maps of differences for Full Air – No Air for 2020. Left column (A, C, E, G) TOA Radiation. Right column (B, D, F, H) land surface temperature. A,B) Dec–Feb (DJF), C,D) Mar–May (MAM) E,F) Jun-Aug (JJA) G,H) Sept–Nov (SON). Stippled regions are significant differences using an FDR test at 90% confidence.





CRE (Figure 5A), LW CRE (Figure 5B), IWP (Figure 5C), ice number (Figure 5D) and ice effective radius (Figure 5E). The high cloud differences are small (Figure 5F) but also similar and opposite to full contrail effects. There is little significance to

annual TOA radiation changes (Figure 5G). Surface temperature over land cools as contrails are reduced (Figure 5H). This has similar and opposite pattern to the full contrail effects, including warming over the Eastern U.S. and cooling over the Western U.S and in subtropical Africa and Asia, with the largest magnitude change -0.2K. The cooling is due to compensating SW and LW effects that vary by season (Figure 2).

In all these cases, the level of significance is small, indicating that the signal due to COVID19 induced changes in contrails

is smaller than variability in most regions. This makes comparisons to observations difficult. However, recent work (Quaas et al 2021, Climate impact of aircraft-induced cirrus assessed from satellite observations before and during COVID-19, submitted to Environ. Res. Lett.) found that in the regions of highest air traffic density there was a 9% decrease in expected cirrus cloud fraction in 2020. An analysis of MAM for high cloud coverage (Figure 5F) shows decreases in cirrus coverage up to 4–5%, smaller than but consistent with observations.

### 3.3   Cloud Changes and Effects of Temperature Nudging

The spatial (Figure 3F) and temporal (Figure 2F) pattern of high cloud changes due to contrails shows significant effects away from flight routes. The vertical structure of the cloud changes, along with temperature and ice water path are shown in Figure 6. Aviation water vapor causes increases in cloud ice concentrations (Figure 6C). This can increase or decrease cloudiness (Figure 6A) depending on the temperature (and humidity) response. Without temperature nudging, there is local

warming due to LW absorption by cloud ice (Figure 6B). The increase in temperature and cloud ice (Figure 6C), some of which comes from forming contrails in supersaturated air and subsequent uptake of environmental water, results in decreasing relative humidity (not shown), and hence decreases cloud fraction (Figure 6A).

Nudging temperature alters the cloud response (Figure 6D), with a larger decrease in clouds at mid-latitudes, and reduced increases in cloudiness in the subtropics. The cloud ice mass response is similar with (Figure 6C) or without (Figure 6F)

temperature nudging. The high cloud increases near the equator are in the sub-tropics and mostly zonal (Figure 3F) and may be associated with temperature increases allowing more water vapor and then cloud ice to be present. This highlights the subtle challenges of describing a radiative forcing, and why we use Effective Radiative Forcing (ERF), including these local temperature responses. COVID19 changes in clouds and ice without (Figure 6 G, H, J) and with (Figure 6 K, L, M) temperature nudging are similar to Full aviation effects but smaller and of opposite sign.

### 4   Discussion and Conclusions

These simulations of aviation contrails and the effect of COVID19 induced reductions provide some interesting new perspectives on the effect of contrails on climate. Contrail Effective Radiative Forcing (ERF) is estimated at $62\pm59$ mWm$^{-2}$ ($2\sigma$) for current (2020) aviation in the absence of any COVID19 pandemic caused reductions. The variability range is due to ensemble spread and differences in year to year meteorology, indicating that the value may vary from year to year. A more complete

**Figure 5.** Annual mean maps of differences for COVID – Full Air for 2020. A) Shortwave (SW) Cloud Radiative Effect (CRE), B) Longwave (LW) CRE, C) Ice Water Path, D) Cloud Top ice number concentration, E) Cloud top ice effective radius, F) High cloud fraction, G) Net Top of Atmosphere (TOA) radiation difference and H) Land surface temperature difference. Stippled regions are significant differences using an FDR test at 90% confidence.



**Figure 6.** Annual zonal mean latitude height plots differences in Cloud Fraction (Left column: A,D,G,K), Temperature (T, Center column: B, E, H, L) and Cloud Ice Mixing Ratio (Right column: C, F, J, M). COVID - full air (Top row: A,B,C), No aviation- full air (2nd row: D, E, F), No aviation - full aviation T nudged (3rd row: G, H, I), COVID - full aviation with T nudging (bottom row: K, L, M). Stippled regions are significant differences using an FDR test at 90% confidence.



analysis of inter-annual variability should be conducted, but is beyond the scope of this study. This range is well in line with recent assessments of contrail ERF (Lee et al., 2021).

The net contrail ERF has complex spatial and seasonal patterns and is a residual of SW cooling and LW heating components nearly 10 times the net effect. The patterns are important to understand, and a significant complexity for assessing aviation contrail impacts. This arises because of the seasonality of SW radiation (Figure 2A) mapped onto the seasonality of flights

(Figure 1), which results in net contrail cooling from June to September (Figure 2G and Figure 4E), but strong global (Figure 2G) and regional (Figure 4A) heating effects in December–February. This is an underappreciated part of the contrail ERF, and may have implications for mitigation strategies.

The spatial and seasonal forcing variations also map onto land surface temperature variations, resulting in less annual temperature change than might be expected. In Western Europe for example, peak net radiation differences occur in fall and winter

when radiation is less a part of the surface energy budget, so little temperature change results. In Eastern N. America, SW cooling effects of contrails dominate in spring and summer, while LW effects occur more strongly in fall and winter, such that there is little annual temperature change. Largest temperature changes are found over subtropical Africa, away from flight routes, but perhaps affected by high cloud increase in these regions due to remote effects of aviation.

The temperature changes resulting from these small ERF's are much smaller than climate variability in a fully coupled

system. For this study is that ocean temperatures are fixed, which is fine on the short term and for small ERF estimates. With this caveat, there are significant increases in temperature over most land regions due to contrails, with an annual average over land of of 0.13±0.04 K. The peak annual temperature change is 0.7K in the N. Hemisphere sub-tropics.

The effect of COVID19 reductions in flight traffic (Figure 1D) decreased contrails. The unique timing of such reductions, which were maximum in N. H. summer when the largest contrail cooling occurs, means that warming reductions due to fewer

contrails in the spring and fall were offset by cooling reductions due to fewer contrails in summer to give no significant annual averaged ERF from contrail changes in 2020. Despite no net significant global ERF, there are some land regions that cooled significantly up to -0.2K from what would have been expected with baseline aviation contrails. These reductions occurred in the same regions as large contrail temperature changes in the subtropical N. Hemisphere.

The patterns of surface warming and cooling are not exactly coincident with contrail ERF, indicating distributed effects

through the climate system. These effects should be further tested not just with a coupled land surface (as has been done here, but with a coupled ocean. However, this will introduce additional climate noise as well, so is a subject for future work.

This study provides estimates based on unique and detailed modeling frameworks to elucidate small changes to the climate system with ensembles of constrained simulations. One important question is whether any of these simulated changes due to COVID19 aviation reductions can be seen in observations. The pattern and timing of radiation and warming changes might

yield sufficient fingerprints in instrumental records of anomalies during 2020 to be able to tease out these effects, and this an interesting avenue for future research. Preliminary work (Quaas et al, 2021, Climate impact of aircraft-induced cirrus assessed from satellite observations before and during COVID-19, submitted to Environ. Res. Lett.) indicates observed decreases in cloud fraction in 2020 in high traffic regions consistent with these simulations. This is one avenue for comparison of these





simulations to observations. Other analyses are possible, and simulation results are available to the community for further
analysis.

What does this analysis mean for the future climate impact of contrails? The cancellation between LW and SW indicates
that the spatial and seasonal distribution of flights may change the contrail ERF. Local effects in space and time may not be the
same as global impacts due to the timing of contrails and solar insolation. If flights increase in tropical regions where there is
more SW radiation throughout the year, this might decrease the ERF of contrails (more cooling). But it also may mean more
flights in regions susceptible to contrails such as the upper troposphere (through regions of ice supersaturation). An updated
aviation emissions data base (more recent than the scaled 2006 ACCRI inventory used here) and projections would be useful
to begin these assessments. The results here also indicate that the seasonal cycle could be used as a contrail mitigation strategy,
whereby one would NOT want to alter or reduce contrails in regions and during times of year with larger SW cooling.

*Code and data availability.*

Simulation output and modified code used in this analysis is available at http://doi.org/10.5281/zenodo.4584078 . Simulations are based on CAM6.2: https://github.com/ESCOMP/CAM/tree/cam6_2_022.

*Author contributions.*  AG Designed the study, did the main simulations and analysis and wrote the manuscript. CCC modified the code, did preliminary simulations and analysis and helped edit the manuscript, and CGB assisted with data sets and editing the manuscript.

*Competing interests.*  None

*Acknowledgements.*  The National Center for Atmospheric Research is funded by the U.S. National Science Foundation. Thanks to flightradar24 for access to total flight data. Thanks to D. S. Lee for discussion and analysis of aviation emissions growth.



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
