# Peer review of "The Climate Impact of COVID19 Induced Contrail Changes"

_Atmospheric Chemistry and Physics, 2021_

## Referee Comment (RC2)

This paper is aimed at analyzing the effects of the reduction in flights during the pandemic period in 2020 to examine the effect on resulting contrails. The NCAR model used in the study has been previously evaluated for analyses of contrails in previous years. The analyses are interesting and have scientific value. The paper is well written and should be publishable after revision.

I have two major comments:

Although the contrail derivation capabilities used in the model were evaluated in much earlier papers (from 2012 and 2013), an update on the capabilities are warranted relative to observations and analyses of contrails that have happened since those papers were published. A complete reanalysis is not required but perhaps as much of a page update on the capabilities of the contrail derivation from the model relative to the literature would be very helpful.

On page 4, line 75: Please justify the lack of considering aviation aerosol impacts on contrails in the study. If they had been included, how much of an impact could they have had on the results.

---

## Author Response (AR1)

>> We thank both reviewers for their time. We have made all the revisions suggested. Most substantially, we have added further justification for not including aerosols, and further analysis of the updated model compared to previous work (including recent observations). We have also made all the minor modifications suggested.

Reviewer #1

Gettelman et al. performed a modelling study to investigate the impact of aviation contrails and aviation water vapour emissions on climate, with a special focus on the effects due to the flight reductions during the COVID-19 pandemic. The study is very timely, since there is currently a large interest in understanding the climate impacts of various consequences of the COVID pandemic, and air traffic reduction is a particularly strong signal. Two caveats on the study are that (i) the impacts of contrail cirrus are neglected (but the argument put forward by the authors that these are uncertain in general circulation models is a convincing one) and (ii) that it is purely a modelling study, with no direct observational evidence (but the authors are of course right that an analysis of observations is not straightforward). Nevertheless, it remains a useful study that is of interest to the readers of Atmos. Chem. Phys. The study is very well written and the results are clearly presented and discussed. I recommend that the paper is published after a few minor revisions.

>> We thank the reviewer for their detailed review. One clarification: contrail cirrus are NOT neglected. This is stated specifically in the model description. It is the effects of aviation aerosols on subsequent clouds that are neglected. We clarify this a bit when the model is discussed to make it clearer that contrails evolve into contrail cirrus. We hope this clarifies.

l50 (and later) better use "$\mu$m" rather than "microns"

>> Corrected

l59 I propose to write "0.1\%" instead of 1.e-4.

>> Corrected

What is the lifetime of the contrail?

>> It varies, from 30 minutes (one timestep) to many hours. Noted.

l65 Is there an interactive ocean? Or what does it mean, that SST are "relaxed" to Merra?

>> Not an interactive ocean, fixed ocean surface temperatures. Clarified.

l98 A brief motivation on why not daily emission change factors (since the weekly cycle is rather well known by Fig. 1A) would be useful.

>> Noted now in the text. It was considered that trying to remove it and reimpose a specified weekly cycle during the pandemic was difficult. Figure 1A indicates that the weekly cycle was altered. That level of detail is well within the noise of these calculations.

l139/Table 1 IWP should have units of g m-2, not g kg-1. Is the cloud fraction indeed in percent, or a fraction as in Fig. 2?

>> IWP Corrected. Clouds: it is fraction. Corrected.

l142 Add the units to the ratio (W m-2 / (kg m-2) = s-3 )?

>> Actually, W = kg·m2·s−3, so Wm-2 = kg s-3 and the ratio is m2/s-3 (Which is a bit strange). Since the units are not really important (and I used grams too, so that would complicate it), the main point is that the relationship does not change. So we have decided to leave the units off.

l160 "Are in equilibrium" is a debatable statement. (a) the SSTs are imposed and so it is anyway unclear how well these correspond to what the model thinks full aviation effects are, and (b) the ocean has much longer timescales than rapid evolution of the transient aviation forcing.

>> Changed to "Since the ocean adjustment time to any forcing is long and the perturbations are much smaller than the variability of radiative forcing, any imbalance due to fixed ocean temperatures should be a small effect well within the uncertainty range of the ensemble."

l208 But in the model, contrail cirrus are not considered so the fact that the simulate decrease is smaller is indeed expected, right?

>> As noted, contrail cirrus ARE considered in the model.

l213 For the ice number, are the contrails or is indeed the water the main effect?

>> This should say 'mass concentrations': It is water vapor. The figure is not discussing number. Corrected.

l245 superfluous "is"

>> Corrected.

Review #2

Review of Gettleman et al.

This paper is aimed at analyzing the effects of the reduction in flights during the pandemic period in 2020 to examine the effect on resulting contrails. The NCAR model used in the study has been previously evaluated for analyses of contrails in previous years. The analyses are interesting and have scientific value. The paper is well written and should be publishable after revision.

I have two major comments:

Although the contrail derivation capabilities used in the model were evaluated in much earlier papers (from 2012 and 2013), an update on the capabilities are warranted relative to observations and analyses of contrails that have happened since those papers were published. A complete reanalysis is not required but perhaps as much of a page update on the capabilities of the contrail derivation from the model relative to the literature would be very helpful.

>> We have added a summary paragraph on further analysis conducted of the contrail simulations in section 2.1, and refer back to this now in the discussion. The key point is a recent paper by Schumann et al. (2021) has enabled us to look at comparisons between 2020 and 2019 in the simulations, and they get similar signs to observed cloud changes.

> "We have compared the results to previous contrail simulations with this model and others, as well as with observations. The pattern of resulting contrail changes illustrated below to cloud fraction are very similar to the previous model documented in CAM5 (Chen and Gettelman 2013) with peak effects in Northern Hemisphere mid-latitudes. The radiative forcing of the CAM6 simulations (discussed in detail below) is larger than in Chen and Gettelman (2013) due to (a) smaller (100m v. 300m) initial contrail area and (b) smaller initial ice crystal sizes (7.5 v 10 μm diameters). The radiative forcing pattern and magnitude matches Bock and Burkhardt (2019), Figure 5, qualitatively and quantitatively. This is consistent with the intercomparison between contrail simulation models that was recently conducted as part of the review by Lee et al (2021). The CAM6 contrail model results also compare well to observations and simulations of contrails by Schumann et al (2021), who analyzed differences in clouds between 2020 and 2019. The results here have the same sign of cloud changes as observed and simulated in Schumann et al (2021). This yields further confidence in the model CAM6 model estimates presented here. "

On page 4, line 75: Please justify the lack of considering aviation aerosol impacts on contrails in the study. If they had been included, how much of an impact could they have had on the results.

>> We have added a paragraph to the discussion as suggested:

> This study has not considered any impacts of changes to aviation aerosol emissions, largely sulfate (SO4) and Black Carbon (BC). Aviation aerosols are highly uncertain, which is why we chose to focus on aviation water vapor. Note that aviation aerosols affect only subsequent cloud formation, not initial contrails and contrail cirrus. In CESM, aviation aerosols, especially SO4, tend to mix downward to affect liquid clouds below (Gettelman et al., 2013). The net effect of the aviation aerosols is a cooling, so COVID reductions would likely cause a net warming. The seasonality is similar to SW effects here. But these effects have not been included because there is a wide divergence in outcomes depending on the model and the background state. These mechanisms have yet to be confirmed and are not quantified even in recent assessments (Lee et al., 2021).